# Major features of parasite adaptation revealed by genomes of *Plasmodium falciparum* population samples archived for over 50 years

Alfred Amambua-Ngwa [1,2,3] ✉, Mouhamadou Fadel Diop [1], Christopher J. Drakeley [2], Umberto d'Alessandro[1], Dominic P. Kwiatkowski [3,4] & David J. Conway [2] ✉

Understanding evolution of human pathogens requires looking beyond the effects of recent interventions. To study malaria parasites prior to widespread drug selection, *Plasmodium falciparum* genomes were sequenced from the oldest population-based set of archived research samples yet identified, placental blood collected in the Gambia between 1966 and 1971. High-quality data were obtained from 54 infected samples, showing that genomic complexity within infections was high, most infections were genetically unrelated, and no drug resistance alleles were detected. Strong signatures of positive selection are clearly seen at multiple loci throughout the genome, most of which encode surface proteins that bind erythrocytes and are targets of acquired antibody responses. Comparison of population samples obtained over a following period of almost 50 years revealed major directional allele frequency changes at several loci apart from drug resistance genes. Exceptional changes over this time are seen at *gdv1* that regulates the rate of parasite sexual conversion required for transmission, and at the unlinked *Pfsa1* and *Pfsa3* loci previously associated with infection of individuals with sickle-cell trait. Other affected loci encode surface and transporter proteins warranting targeted functional analyses. This identification of key long-term adaptations is important for understanding and managing future evolution of malaria parasites.

Population genomic studies of malaria parasites have yielded key insights on selection operating under different malaria control contexts. These have helped understanding of drug resistance emergence and spread, to inform antimalarial treatment and chemoprevention policies[1–7]. However, it is important to understand also the modes of selection that operated prior to drug resistance emergence, and which continue to affect interactions between parasites and their hosts[8]. Ancient DNA from excavated remains of infected individuals have begun to give insights on past evolution of malaria parasites[9], and museum specimens may also contain parasite DNA for analysis[10,11], but such sources have yielded only sporadic positive samples.

Early research archives containing sufficient material to enable population-based analyses on historic selective processes are particularly valuable. Previous analyses of archived population samples of *Plasmodium*

*falciparum* taken at different times in The Gambia dating back to the 1980s have revealed the local history of drug selection[1,12–14]. Chloroquine resistance started to appear in The Gambia from the late 1980s onwards due to selection for variants of the *aat1* and *crt* drug transporter genes[1,14]. Following this, the antifolate combination sulphadoxine-pyrimethamine was introduced as second-line treatment during the 1990s and was added to chloroquine as first line treatment in 2004, by which time antifolate resistance had already increased due to selection of *dhfr* and *dhps* variants encoding altered drug target sites[1,14]. Artemisinin-based combination therapy was introduced in 2008, although malaria incidence in The Gambia started declining slightly earlier[15–19]. Besides drug resistance, genomic studies on parasites in The Gambia have highlighted the operation of balancing selection on targets of immunity[20], and allowed comparison of local signatures of selection with those occurring in other parasite populations in Africa[13,21,22].

[1]MRC Unit The Gambia at London School of Hygiene and Tropical Medicine, Banjul, The Gambia. [2]Department of Infection Biology, London School of Hygiene and Tropical Medicine, London, UK. [3]Wellcome Sanger Institute, Hinxton, UK. [4]Deceased: Dominic P. Kwiatkowski ✉e-mail: alfred.ngwa@lshtm.ac.uk; david.conway@lshtm.ac.uk

Aiming to understand selection on malaria parasites in the more distant past, and over a longer period, records of earlier studies were examined to seek the oldest archived samples containing parasite material. These were found to be placental blood samples collected in The Gambia in 1966–1971, which originally enabled the first analysis of *P. falciparum* isoenzyme variation[23] and description of polymorphic antigens[24,25], undertaken in the era before methods to analyse DNA were available, and long before any antimalarial drug resistance was seen locally. Here, the parasite genome sequences from the entire available set of samples are analysed, revealing the key features of positive selection that had been operating on parasites before emergence of drug resistance. More recent samples from the same area were then added to enable the longest temporal analysis yet performed, which identified new targets of selection occurring over the subsequent period of five decades. This identifies ways in which parasites continue to evolve and adapt to their human hosts, highlighting targets for development of future tools to sustain malaria control in future.

## Results

### Identifying targets of selection on malaria parasites in an era prior to drug resistance

Archived *P. falciparum*-positive placental blood samples collected in 1966–1971 from the coastal area of The Gambia were identified and processed for parasite genome sequencing. Fifty-four (90%) of 60 samples yielded >80% genome-wide coverage at a read depth of at least 5x (Supplementary Data 1), enabling a powerful analysis of genome-wide diversity.

Initially, the genome sequences of the 54 infections were considered together as a population sample from this era, enabling analysis of diversity throughout the genome. To analyse population-wide *P. falciparum* allele

frequencies for all SNPs, for each infection the predominant genotype profile for each SNP was defined by the majority of sequence reads mapped, and genome-wide tests for positive selection were applied to the profiles from all infections. These data show that strong positive selection was operating on multiple loci in the parasite genome long before drug resistance emerged in this population, and there were no drug resistance loci under selection (Fig. 1 and Supplementary Data 2 and 3).

Twenty-six genes in sixteen separate genomic regions had the highest values of the beta score index which is generally an indication of long-term balancing selection (Fig. 1 and Table 1), and 14 of these genes contained SNPs with significant integrated haplotype scores (iHS), indicating recent positive directional selection on some alleles (Fig. 1 and Table 1). Co-occurrence of these signatures of selection on a gene suggests that balancing selection generally maintains a diversity of alleles while introduction of new alleles to the repertoire may continue to occur.

Most of these parasite genes under positive selection encode surface proteins that are targeted by immune responses. These include merozoite surface proteins (MSP1, MSPDBL2), four members of one family of proteins expressed on merozoites or infected erythrocytes (Surfins 1.1, 4.2, 8.2 and 14.1), three members of another family of surface and rhoptry apical organelle proteins (CLAGs 2, 3.1/3.2, and 8), micronemal apical organelle proteins involved in erythrocyte invasion (EBA140 and AMA1) and a sporozoite surface protein (TRAP). Others encode a less characterised protein with a Duffy binding-like domain (DBL1), as well as uncharacterised proteins predicted to be exported from the intraerythrocytic parasite. Most genes with these signatures of selection are in or close to chromosomal subtelomeric regions, and are heterochromatic loci that show epigenetic variation in expression[26].

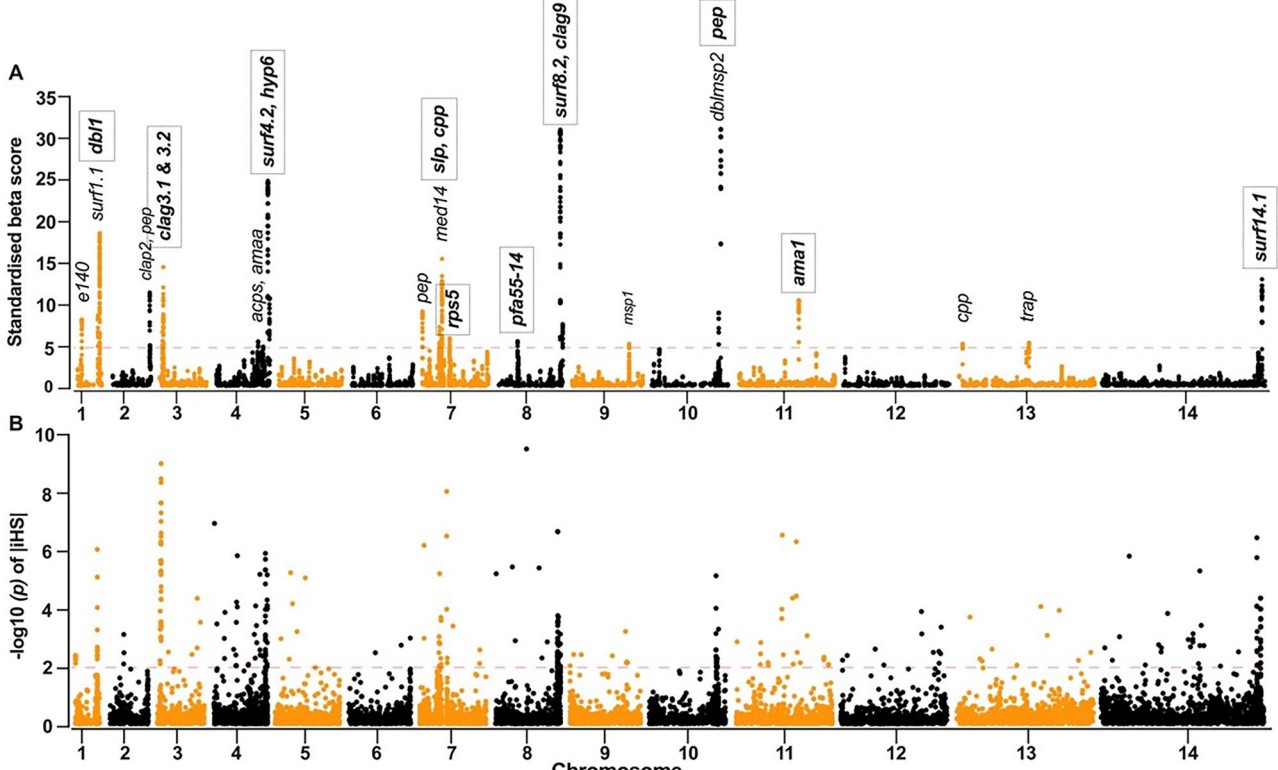

**Fig. 1 | Genome-wide scans for signatures of selection on the *P. falciparum* population in The Gambia sampled in 1966–1971, many years prior to the emergence of drug resistance. A** The beta score index across sliding windows of SNPs for all 14 chromosomes identifies loci with correlated SNP allele frequency distributions indicating balancing selection. Twenty-six genes in the windows with the highest beta scores are labelled, and 14 of these (presented in bold font within boxes) also contained at least one SNP with a significant integrated haplotype score

(iHS)(Table 1). **B** The -log10 *p* value for the integrated haplotype score (iHS) which indicates recent directional selection on the basis of extended haplotype homozygosity is shown for each SNP. Most of the genes showing these signatures of positive selection encode parasite surface proteins that interact with host cells and are targets of antibody responses. Three genes with the label '*pep*' and two as '*cpp*' have no specific name or putative function (all gene IDs are shown in Table 1). Values for beta score and iHS indices of all genes analysed genome-wide are given in Supplementary Data 2 and 3.

**Table 1 | Population-based genome-wide scan of 54 archived *P. falciparum* infection samples from 1966–1971 identifies 26 genes with strong signatures of positive selection in an era prior to the emergence of drug resistance in The Gambia**

| Gene Symbol | Gene ID | Product Description | Number of base pairs in coding sequence | Number of SNPs | Beta Score mean | iHS -log10 P-value max | Tajima's D value |
|---|---|---|---|---|---|---|---|
| E140 | PF3D7_0104100 | protein E140, putative | 1464 | 45 | 7.89 | 0.67 | 0.83 |
| SURFIN1.1 | PF3D7_0113100 | surface-associated interspersed protein 1.1 | 4665 | 49 | 7.94 | 1.72 | 0.15 |
| DBI-1 | PF3D7_0113800 | DBL containing protein, unknown function | 1143 | 144 | 13.69 | 6.19 | 0.82 |
| CLAG2 | PF3D7_0220800 | cytoadherence linked asexual protein 2 | 4320 | 39 | 9.97 | 1.86 | 2.15 |
| PEP | PF3D7_0221000 | Plasmodium exported protein, unknown function | 906 | 20 | 5.28 | 1.47 | 1.31 |
| CLAG3.2 | PF3D7_0302200 | cytoadherence linked asexual protein 3.2 | 4248 | 46 | 5.99 | 4.00 | -1.38 |
| CLAG3.1 | PF3D7_0302500 | cytoadherence linked asexual protein 3.1 | 4251 | 81 | 8.80 | 9.23 | 0.56 |
| ACPS | PF3D7_0420200 | holo-[acyl-carrier-protein] synthase, putative | 2205 | 13 | 5.49 | 0.56 | 2.29 |
| EMAA | PF3D7_0422200 | erythrocyte membrane-associated antigen | 12783 | 18 | 5.11 | 1.26 | -1.02 |
| SURFIN4.2 | PF3D7_0424400 | surface-associated interspersed protein 4.2 | 7140 | 129 | 20.85 | 6.05 | 0.73 |
| Hyp6 | PF3D7_0425100 | Plasmodium exported protein (hyp6), unknown function | 615 | 30 | 9.87 | 4.40 | 2.38 |
| PEP | PF3D7_0701900 | Plasmodium exported protein, unknown function | 2892 | 53 | 8.05 | 0.65 | 1.61 |
| MED14 | PF3D7_0709300 | mediator of RNA polymerase II transcription subunit 14, putative | 8187 | 33 | 6.20 | 1.86 | 0.2 |
| SLP | PF3D7_0710000 | Sfi1-like protein SLP | 9801 | 36 | 5.83 | 3.10 | 0.65 |
| CPP | PF3D7_0710200 | conserved Plasmodium protein, unknown function | 8730 | 89 | 10.87 | 5.33 | 0.88 |
| RPS5 | PF3D7_0713600 | ribosomal protein S5, mitochondrial, putative | 585 | 20 | 5.88 | 2.34 | 0.7 |
| pfa55-14 | PF3D7_0809200 | asparagine-rich antigen Pfa55-14 | 3948 | 23 | 5.64 | 2.95 | 0.29 |
| SURFIN8.2 | PF3D7_0830800 | surface-associated interspersed protein 8.2 | 6147 | 149 | 24.69 | 6.82 | 1.47 |
| CLAG8 | PF3D7_0831600 | cytoadherence linked asexual protein 8 | 4182 | 44 | 6.92 | 3.19 | 0.54 |
| MSP1 | PF3D7_0930300 | merozoite surface protein 1 | 5160 | 89 | 5.18 | 2.18 | 0.78 |
| PEP | PF3D7_1035100 | probable protein, unknown function | 1683 | 37 | 8.38 | 2.39 | -0.02 |
| DBLMSP2 | PF3D7_1036300 | duffy binding-like merozoite surface protein 2 | 2286 | 80 | 28.97 | 1.26 | 2.86 |
| AMA1 | PF3D7_1133400 | apical membrane antigen 1 | 1866 | 40 | 9.83 | 6.46 | 1.78 |
| CPP | PF3D7_1302900 | conserved protein, unknown function | 2400 | 13 | 5.33 | 0.64 | 1.88 |
| TRAP | PF3D7_1335900 | thrombospondin-related anonymous protein | 1677 | 40 | 5.52 | 1.01 | 1.22 |
| SURF14.1 | PF3D7_1477600 | surface-associated interspersed protein 14.1 | 5859 | 68 | 10.97 | 4.46 | 0.91 |

Signatures of positive selection are indicated by the mean beta score for each gene, listed alongside significance of integrated haplotype score (iHS) and Tajima's *D* value. Genome-wide data for the tabulated indices are given in Supplementary Data 2–4. Three genes indicated with the gene symbol 'PEP' (Plasmodium exported protein) and two as 'CPP' (conserved Plasmodium protein) have no specific name or putative function. Gene coding sequences refer to the 3D7 reference genome.

**Fig. 2 | Genomic complexity within *P. falciparum* infections sampled in The Gambia in 1966–1971 and in 2015 (shown in red and blue points respectively). A** Within-infection fixation index ($F_{WS}$) which is an inverse measure of genomic complexity (values of 1.0 corresponding to single-genome infections), with each infection represented by a separate point. Mean values for each sampling period are shown by dashed horizontal lines. The median $F_{WS}$ values were 0.74 for 1966–1971 and 1.0 for 2015. **B** 'Complexity of Infection' (COI) estimations using the REAL McCOIL method. The differences between sampling periods were highly significant ($P < 0.0001$), indicated by asterisks ****. The values of $F_{WS}$ and COI for the individual infection samples in 1966–71 and 2015 are given in Supplementary Data 5 and 7 respectively.

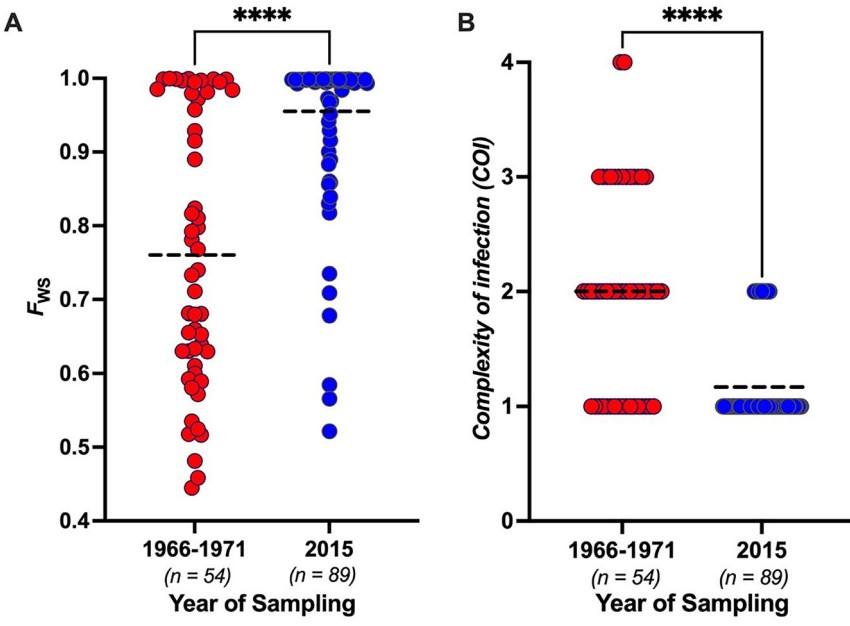

Using another approach to identify genes with alleles at more intermediate frequencies than expected, a scan of Tajima's *D* values was performed on the genome-wide data from the 1966 to 1971 population (Supplementary Data 4, Supplementary Fig. 1). Twenty three of the 26 genes with highest beta scores also had positive values of Tajima's *D* (Table 1 and Supplementary Fig. 1). As absolute values of Tajima's *D* are generally subject to demographic effects, it is only loci with exceptional values compared to the rest of the genome that are noted as potentially influenced by selection. Most of the genome has negative Tajima's D values (Supplementary Fig. 1) consistent with hypothetical long-term population expansion, Therefore, against this genomic background the positive values in Table 1 are extreme. In many cases this is likely to be due to frequency-dependent selection exerted by the strong memory component of human B and T cell responses, as previously noted for known targets of immunity[20,27,28].

### Genomic complexity of infections compared to more recent population samples

Many of the individual infection samples from 1966 to 1971 contained significant *P. falciparum* genomic sequence diversity, indicating that these contained multiple haploid parasite genotypes. The within-infection fixation index ($F_{WS}$ which has a value of less than 1.0 when more than one genotype is present) had values of < 0.95 for 38 (70%) of the 54 isolates (Fig. 2A, Supplementary Data 5), indicating that most infections contained multiple different parasite genotypes. The overall mean $F_{WS}$ value was 0.76 (median of 0.74), much lower than 1.0 and driven by particularly infections containing high levels of parasite genomic diversity. In recent population samples from across Africa, such low $F_{WS}$ values are seen only in areas with very intense malaria transmission[29,30].

To analyse the genomic diversity of infections from the same coastal area of The Gambia almost five decades later, parasite genomic DNA was extracted from blood samples taken from patients with uncomplicated malaria attending Brikama Hospital in 2015, and sequenced for analysis using the same methods used for the oldest archived samples (Supplementary Data 6). Within-infection diversity among 89 samples collected in 2015 was much lower, with most $F_{WS}$ values close to 1.0 and an overall mean $F_{WS}$ value of 0.96 and median of 1.0 (Mann-Whitney test comparing the sampling periods, *P < 0.0001*) (Fig. 2A, Supplementary Data 7). Consistent with this, using a different approach to estimate complexity of Infection (COI) as the minimum number of distinct genotypes detectable in each

sample, the COI levels were much lower in 2015 than in 1966–1971 (Fig. 2B, Supplementary Data 7)(*P < 0.0001*).

### Population-wide genomic diversity compared to more recent population samples

To further compare the population samples, the extent of genomic differences among different infections was analysed. For this purpose, the dominant *P. falciparum* profile for each infection was considered, represented by SNP variants in the majority of mapped sequence reads, using a subset of genome-wide SNPs with no missing genotypes and low pairwise linkage disequilibrium. The population-wide diversity between infections was high among the archived samples from 1966 to 1971, and also among the more recent samples from 2015 (Fig. 3). Visualisation of overall diversity patterns by multidimensional scaling analysis revealed slightly more population structure among the 2015 samples, including a minority of infections being outliers (Fig. 3A). Hierarchical clustering of pairwise similarity between the genomic profiles of different infections shows a wide range, as expected from recombination resulting from sexual reproduction that occurs during transmission by mosquitoes (Fig. 3B). A minority of infections showed a high level of inter-relatedness, although this was particularly rare among the older samples, as only two pairs of samples from 1966 to 1971 showed SNP profiles with >95% identity whereas there were 26 such pairs in 2015.

### Identifying parasite loci with exceptional allele frequency changes over time

To identify which regions of the *P. falciparum* genome have undergone the most pronounced allele frequency changes since 1966-1971, a comparison with the 2015 data was first performed (Fig. 4A, Supplementary Data 8). By applying the temporal $F_{ST}$ fixation index, it is clear that discrete loci have undergone marked changes in frequency, whereas the genome-wide background shows minimal frequency change. Four of the top ten genomic loci with high fixation indices contain drug resistance genes (*dhfr*, *aat1*, *crt*, and *dhps*) already known to have been under selection in The Gambia[1,14]. There were no drug resistance alleles at any of these loci in the 1966-71 samples, but these became common from the 1980s onwards (Supplementary Data 9, Supplementary Fig. 2), as has been previously described in refs. [1,14]. Different processes of selection have operated on the other genomic loci showing exceptional levels of allele frequency change over time (*pfsa1*, *gdv1*, *pfsa3*, *msp7*, *mfs6* and *cpp-5*).

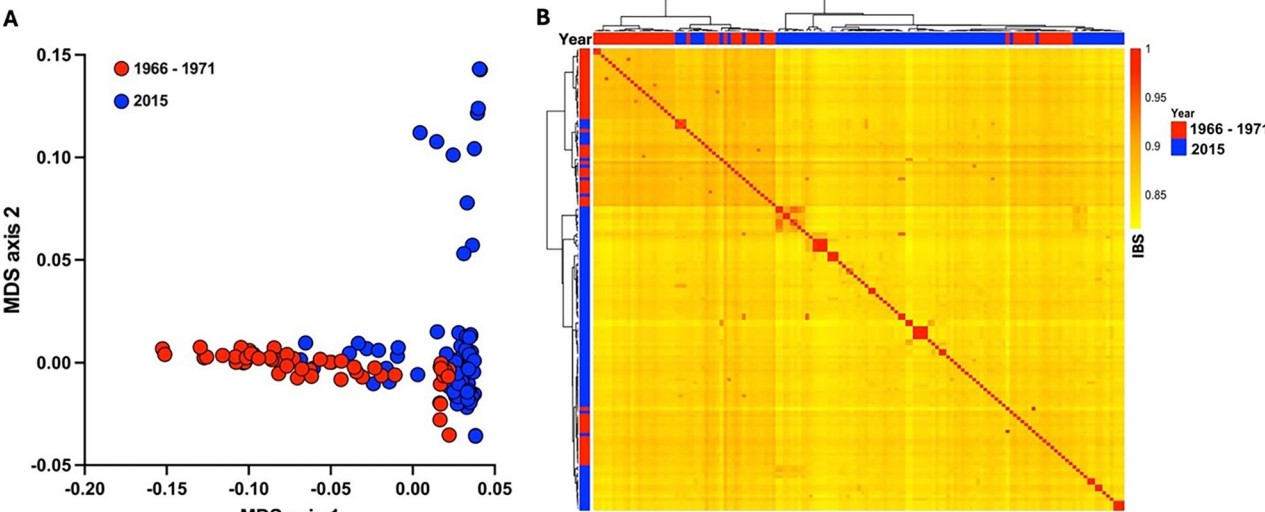

**Fig. 3 | Population-wide diversity of *P. falciparum* genomic profiles in The Gambia in 1966–71 (*N* = 54, shown in red) and 2015 (*N* = 89, blue). A** The genomic profile diversity of infections is shown as a scatter plot of axes 1 and 2 of multidimensional scaling (MDS) of the eigenvectors of distances, based on the SNP profile of each of the infection samples, considering the predominant SNP at each locus within each individual infection. Analysis is based on 2822 genome-wide SNPs that do not have significant linkage disequilibrium. **B** Neighbour-Joining dendrogram-clustered heatmap of pairwise SNP identity-by-state (IBS) between infections.

A minority of pairs of infections showed high relatedness, particularly among the 2015 samples, identifiable by red squares along the diagonal of the IBS heatmap. Overall, the distributions of infection samples from the different eras were overlapping and interspersed, although some isolates appeared to cluster together for each era, including 11 samples from 2015 with high values on the MDS axis 2 in (**A**) (also clustering in Panel B within a block shaded blue near the middle of the visual matrix). The accession numbers for all the sequence data are given in Supplementary Data 1 and 6.

Analysis of allele frequencies in additional population samples collected at intervening time points in The Gambia (in 1984 and 2001) allow the long-term temporal trajectories to be more fully examined (Fig. 4B, Supplementary Fig. 3 and Supplementary Data 9). Besides the drug resistance-associated loci, the gene with the highest temporal differentiation is *gdv1* (PF3D7_0935400 on chr 9) that regulates parasite sexual commitment[31,32] and controls expression of the MSPDBL2 antigen in mature schizonts[33]. The most temporally-differentiated polymorphism in the *gdv1* gene (SNP causing the amino acid change P217H), increased in frequency from less than 20% to almost 90% (Fig. 4B). The temporal data also showed allele frequency change at some closely-linked loci (Supplementary Fig. 3 and Supplementary Data 9), including the *clag9* gene (PF3D7_0935400, Supplementary Fig. 4).

Notably, two of the other loci showing marked frequency change over time are likely to affect parasite replication in specific human hosts. Variation at the *Pfsa3* locus on chr 11 has been associated with parasite infection of individuals with sickle-cell trait (carrying the HbS allele that protects against severe malaria)[8]. The most temporally differentiated polymorphic site within this chromosomal locus is codon K76E in gene PF3D7_1127000 (Fig. 4B) encoding a putative tyrosine phosphatase observed in the parasite food vacuole[34], while the codon D7V polymorphism in the same gene shows almost the same level of temporal change (Supplementary Fig. 3). Remarkably, the *Pfsa1* locus on chr 2 has also been associated with infections of individuals having sickle-cell trait, and appears to be under parallel selection, showing strong linkage disequilibrium with *Pfsa3* although it is physically unlinked[8]. At the core of the *Pfsa1* locus are six polymorphic codons in the Acyl-CoA synthetase gene (PF3D7_0215300) showing temporal differentiation at approximately the same level (Fig. 4 and Supplementary Fig. 3).

Another polymorphic site with high temporal differentiation maps to the *msp7* gene (PF3D7_1335100 on chr 13) which encodes a merozoite surface protein, near genes encoding other surface antigens, including the sporozoite surface protein TRAP, and Rh2b involved in erythrocyte invasion which have previously shown significant signatures of selection[35–37]. The *trap* gene (PF3D7_1335900) contains a SNP encoding a D166N codon polymorphism that also shows significant temporal changes over the period analysed (Supplementary Data 9, Supplementary Fig. 4).

The *mfs6* gene on chr 14 encodes a major facilitator superfamily-domain-containing protein which is a membrane transporter[38]. The orthologue of this protein in the rodent malaria parasite *P. berghei* is required for optimal replication of parasites during the blood stage of infection, and essential for enabling a new infection to be established from an infected mosquito bite[39]. Experiments with *P. falciparum* clinical isolates would be needed to test whether the variants of this gene affect intrinsic multiplication rates, which vary significantly among isolates in The Gambia[40] and elsewhere. The other locus with significant temporal change located distally on chr 14 has only previously been denoted as a conserved *Plasmodium* protein (*cpp*) as it has orthologues in all malaria parasite species, but it is predicted by gene ontology to be an integral membrane protein.

## Discussion

Analysis of the oldest archived endemic population samples of *P. falciparum* infections here has revealed the principal targets of positive selection operating on parasites prior to emergence of drug resistance. Furthermore, comparison with more recent samples in The Gambia has uncovered other distinct adaptive changes occurring progressively over several decades in a continuously endemic population.

Originally, the targets of positive selecton were mostly genes encoding parasite surface and secreted proteins that bind to erythrocytes, as well as proteins embedded in the infected erythrocyte membrane. Complementary to these findings, it is notable that genomic studies of humans indicate particularly strong selection by malaria on genes affecting erythrocytes[41,42]. Results here indicate that balancing selection operated on most of the affected parasite genes, and many of these also show signatures of positive directional selection. The co-occurrence of these signatures suggests new alleles may be added over time to the repertoire being maintained by selection, consistent with what would be expected from frequency-dependent selection on antigenic targets by acquired immune responses. It is possible that higher malaria transmission and superinfection rates in the past might have caused more intense immune selection than occurs today, so that additional antigen genes were identified as under selection compared with analysis of more recent population samples in the same country[12]. It is

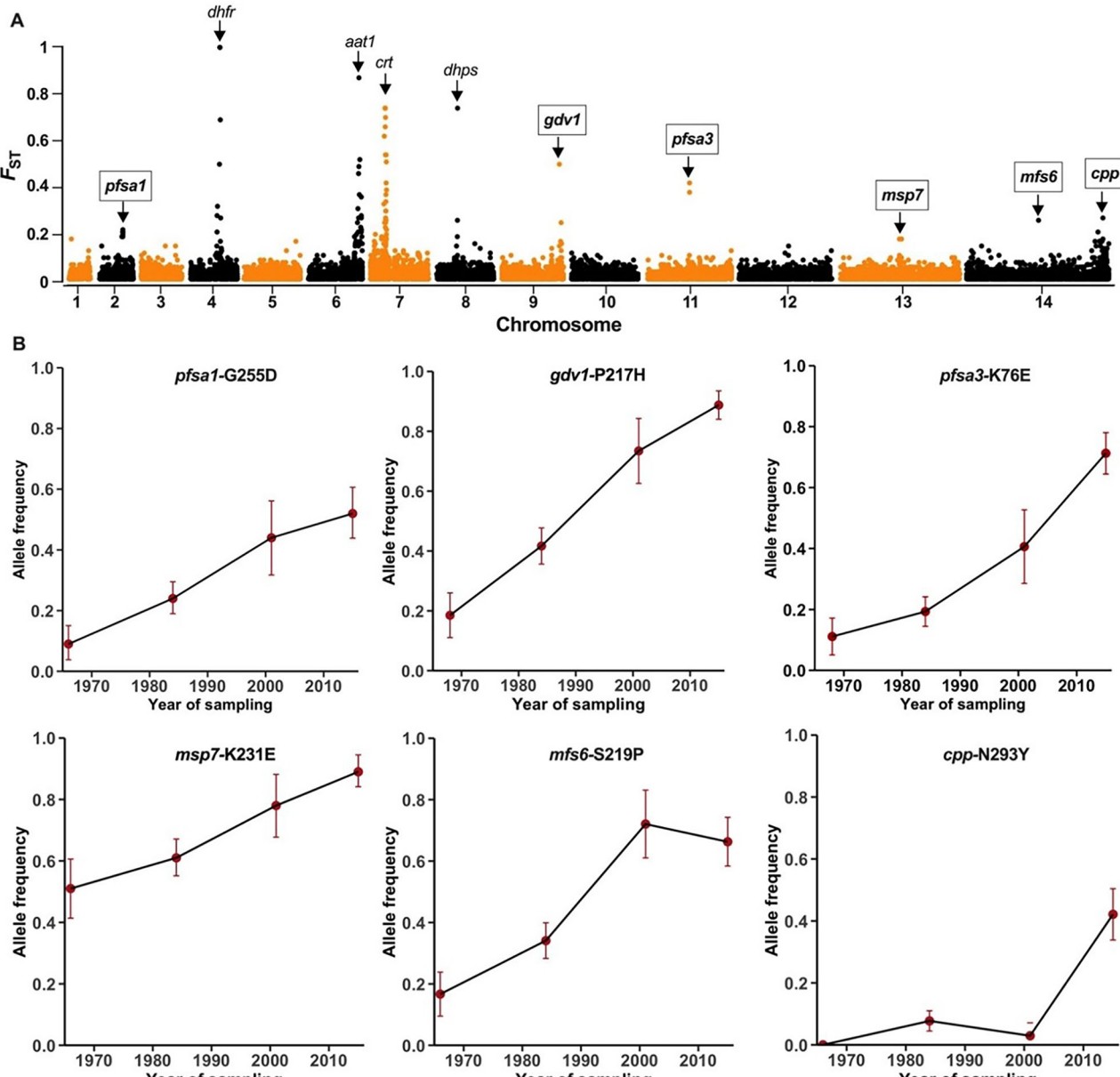

**Fig. 4 | *P. falciparum* genomic polymorphisms with the most pronounced changes in allele frequencies over a period of almost 50 years in The Gambia. A** Genome-wide scan for temporal differentiation shown as $F_{ST}$ temporal fixation indices on SNP allele frequencies in the oldest population samples ($N = 54$ from 1966 to 1971) compared with more recent population samples also sequenced in this study ($N = 89$ from 2015). The plotted genome-wide $F_{ST}$ values are given in Supplementary Data 8. The labels identify the top ten genomic loci showing high temporal allele frequency differentiation. Four of these loci (with unboxed labels) are known drug resistance genes previously shown to be under long-term selection in The Gambia (*dhfr*, *aat1*, *crt*, and *dhps*)[1,14] for which resistance alleles had zero frequency in the 1966-71 samples. The other six loci (with boxed gene labels) showing exceptional frequency change over time are unrelated to drug resistance, and are examined further here. **B** Trajectories of temporal allele frequency changes at

the most differentiated SNPs in these six gene loci. These include data from previously obtained parasite genome sequences[1] from other intervening times (1984 and 2001), alongside the data from 1966 to 1971 (plotted here using the median sample year 1968) and 2015. Allele frequencies at each time point are shown together with 95% confidence interval bars indicating precision based on sample sizes. The frequencies for these and other polymorphisms with temporal $F_{ST}$ values > 0.1 are given in Supplementary Data 9. The *Pfsa1* and *Pfsa3* loci are associated with infection of individuals with sickle cell trait[8]; *gdv1* regulates parasite sexual commitment[32] and expression of a major variant merozoite surface protein[33]; *msp7* encodes a merozoite surface protein[65]; *mfs6* encodes a membrane transporter[39]; *cpp* encodes a conserved *Plasmodium* protein with gene ontology prediction as an integral membrane component.

also possible that the advent of strong selection on drug resistance genes might reduce the ability to detect some signatures of immune selection in recent population samples.

The high level of within-infection *P. falciparum* diversity in the oldest archived samples suggests that malaria transmission and superinfection in The Gambia at the time was intense, consistent with epidemiological reports from that era[18,43]. As the more recent samples from The Gambia show lower

levels of within-infection diversity, it may be speculated that this temporal difference is due to the significant reduction in malaria in the country[15,16,19]. However, it should be noted that the old samples were from placental blood, whereas more recent samples were from peripheral blood of malaria cases, so the comparison is not strictly standardised. Previous studies of placental blood samples have not indicated a higher *P. falciparum* within-infection diversity compared to corresponding peripheral blood samples of pregnant

women[44,45], but comparisons between pregnant and non-pregnant women were not made, so the possibility that placental samples could contain more within-infection diversity than other infections is not excluded.

Analysis of subsequent genomic changes in the parasite population over almost 50 years reveal ongoing adaptions in parasite-host interactions. As expected, drug resistance variants were completely absent from the old archived samples but appeared later in The Gambia and then rapidly increased in frequency, whereas most other significant allele frequency changes involve variants already present in the late 1960s. Among the genomic loci not associated with drug resistance, the strongest evidence of directional selection over time is seen in the gametocyte development gene *gdv1*. This should be considered alongside previous analyses indicating differential selection on this locus in different endemic areas. Notably, the polymorphism with highest temporal change here (SNP causing the codon polymorphism P217H) was previously shown to have the highest geographical differentiation among West African populations[22,46]. The *gdv1* allele 217H that has become frequent in The Gambia is still relatively rare in most other West African populations (except for Senegal which closely neighbours The Gambia), indicating that the selection has operated in this particular far-western area within West Africa[46].

Selection on *gdv1* is likely to reflect adaptation to altered transmission conditions, given its key role in regulating parasite sexual commitment to gametocyte development[32]. Interestingly, in Ghana, where the *gdv1* 217H allele is much less common than in The Gambia[46], it has been reported to be associated with parasites having higher rates of gametocyte conversion in ex vivo culture[47]. Interpretation and further testing requires recognition that the SNP at codon 217 has been shown to be in linkage disequilibrium with other polymorphisms in the coding sequence and in the 3'-intergenic region, as well as several neighbouring genes[46] including the *clag9* gene that encodes a protein with evidence of function in merozoites[48] and infected erythrocytes[49]. Furthermore, the *gdv1* gene also regulates epigenetic expression of the variant merozoite surface antigen MSPDBL2 that is not involved in sexual commitment[33], so identifying the adaptive function of *gdv1* polymorphism may require consideration of more complex phenotypes.

It is remarkable that variants in both the *Pfsa1* and *Pfsa3* loci on chromosomes 2 and 9, respectively, associated with infections of sickle-cell trait individuals[8], show exceptional changes in allele frequencies over time. Previous analysis of *Pfsa3* genotype frequencies in Gambian samples from malaria patients indicated changes in allele frequency between 1999 and 2008[8]. Our results here show that changes have been more profound and continuous over a much longer period. As the frequencies of sickle cell genotypes have not changed significantly in the local human population during this period, we speculate that these parasite variants might also affect the fitness of infections in individuals with different levels of haemoglobin, including those with anaemia from various causes. The mean levels of haemoglobin in children in The Gambia have increased substantially during the study period[50,51], partly due to a reduction of the malaria burden[15,52], and this may have had a selective effect on the parasite[53]. The parasite phenotypes circulating in the population currently, for example having variable multiplication rates[40], are likely due to selective processes operating over an extended period, involving previous interactions with such host conditions.

The approach of identifying and analysing archived biomedical samples can help understand historical and long-term selection. The potential use will depend to some extent on the original methods used for archiving and maintaining research specimens, distinct from the analysis of ancient DNA that may be obtained in smaller amounts from archeological specimens[54,55]. For plant pathogens, this has usually involved dried herbarium samples, which has allowed informative analysis of plant pathogens back to the 19th century[56,57]. For human and animal pathogens, museum samples might contain some tissue material collected in an era prior to biomedical research on pathogens[10,11], although the numbers of such specimens are likely to be too low for analyses of population allele frequencies. The value of having preserving research samples for pathogen analysis is thereby highlighted. As this study has uncovered key processes of selection on a malaria parasite species that still causes over half a million deaths annualy, it illustrates the importance of investing in sample maintenance by institutions in low income countries, as one pillar towards building more equitable research power[58].

## Methods

### Identification of *P. falciparum*-positive archived blood samples and DNA extraction

A search was undertaken for parasite population samples that might be considerably older than those previously sequenced (the oldest samples previously analysed were from 1984)[1]. Examination of archived materials from studies at the MRC Unit in The Gambia identified 61 *P. falciparum*-positive lyophilised placental blood samples from donors sampled in 1966–1971. These were routine deliveries at health centres in the Greater Banjul area near the coast of the country. Donors were not selected on the basis of having any medical condition, and gave informed consent to the placental sampling for analysis of parasites and serum components. Samples were anonymised prior to researcher access. All ethical regulations relevant to human research participants were followed. As these samples were originally analysed in the early 1970s in a study of parasite enzyme electrophoretic variation[23], it was anticipated that most would contain sufficient parasite material to enable selective whole genome amplification for Illumina paired-end short read sequencing. Under approval by the Joint Ethics Committee of the MRC Unit and the Gambian Government, all of these archived samples were processed for analysis by first extracting DNA from the lyophilised blood recovered from a single replicate tube for each sample. The lyophilised blood was effectively solubilised by addition to sample extraction buffer from a QIAamp® Blood Extraction Midi kit (QIAGEN, UK), and DNA was extracted using the same kit with the protocol as previously applied to samples of whole blood.

### Sequencing of *P. falciparum* genomes and SNP variant calling from archived blood samples

Whole genome sequencing was performed using the same methods as applied to recent samples in previous studies[12]. Briefly, selective whole genome amplification to enrich for *P. falciparum* DNA compared to human DNA was performed using a previously described method[59], and paired-end short read (150 bp) sequencing was performed on the Illumina HiSeq platform at the Wellcome Sanger Institute under the MalariaGEN pipeline, which generated parasite sequences for 60 (all except one) of the samples. The sequence data for all of these samples are available in the ENA database (accession numbers and data on read coverage are given in Supplementary Data 1). Variant calling followed established processes that had been used in the MalariaGEN Pf7 pipeline (described for a previous release of global data[5] within which the current data from the 1966 to 1971 infection samples were not included), as in previous analyses of data from The Gambia[1,12].

There were 49,565 coding SNPs detected among these archived samples, of which 48,376 (97.6%) were bi-allelic. Of these, 35,998 had allele calls supported by a minimum of 5 reads and an MQ quality score of at least 30 within at least 80% of the individual infection samples. Following filtration to not more than 20% missingness for variant calls, 54 infection samples were available for the population genetic analyses (the remaining six samples had much lower sequence coverage and were not analysed)(Supplementary Data 1). The exact numbers of SNPs used for each of the different analyses are noted in the relevant sections below.

To enable comparisons with parasites sampled from the same area of The Gambia almost 50 years later, Illumina short-read genome sequencing was performed on 89 *P. falciparum*-infected blood samples from malaria cases presenting to Brikama Health Centre in 2015, male and female children with a mean age of 8 years (range 5 months–14 years). Samples were processed using the same methods, with variant calling performed using the same pipeline to allow analysis of the same set of genome-wide SNPs. Analyses of *P. falciparum* parasite population samples from intervening time periods in The Gambia utilised previously generated sequence data from infections sampled in 1984 and 2001[1,12,14,20], with variant calling being performed in parallel with the other samples for the analysis here.

## Statistics and reproducibility

Details on the statistical analyses, the sample sizes, and the numbers of genome-wide SNPs analysed in each of several different types of tests are given in the following four sub-sections below. Further information on the processing is given in the Github page https://github.com/mrcg-mpb/pfalciparum_50yrs_adaptation_dynamics.

## Genomic complexity of infections and within-isolate fixation indices

For each *P. falciparum* infection sample from which genome-wide sequence data were obtained here (54 from 1966–1971 and 89 from 2015), the degree of genomic mixedness was determined using the within-infection fixation index $F_{WS}$ and an estimate of complexity of infection (COI). $F_{WS}$ was determined using the moimix R package[60] by analysis of 3927 SNPs with no data missingness in any infection sample and a minor allele frequency of at least 5% in both the 1966–1971 and 2015 data. A measure of genomic complexity of infection (COI) that aims to enumerate clearly different genotypes was also generated using the REAL McCOIL method[61] with McCOILR package scripts employed that are available in the Github repository.

## Population structure and pairwise relatedness

Population genotype VCF files were converted to plink using VCFtools. Based on pairwise linkage disequilibrium SNPs were pruned using an $r^2 > 0.1$ as cutoff over a sliding window 20,000 and 10,000 jumps along each chromosome to retain 6267 SNPs with low linkage disequilibrium. From these, a set of 2822 SNPs with no missingness in any sample was used to generate pairwise distances and data reduction by multidimensional scaling (MDS) in PLINK. MDS eigenvectors were plotted as scatter plots in R to visualise clusters. Pairwise identity-by-state (IBS) was calculated and a genetic distance (1-IBS) matrix between all pairs of isolates was displayed as a heatmap using Pheatmap package in R. To position genetically related isolates in the MDS on the IBS-based heatmap, the first three dimensions of MDS were used to generate euclidean distances between isolate pairs and a neighbour joing dendrogram was generated. The topology of the dendrogram was employed in arranging IBS-based heatmap rows and columns for visualisation.

## Identifying genomic regions under selection in The Gambia in 1966–1971

To detect signatures of balancing selection in each population sample we first used the allele frequency-based summary statistic beta score (β), followed by Tajima's *D* index. For Tajima's *D*, 33,034 variants in non-repeat coding regions for 3038 genes with at least 3 coding polymorphic SNPs irrespective of minor allele frequency were analysed and the index was calculated for each gene using custom R scripts as previously described in ref.[20]. The beta statistic was calculated using the python package BetaScan, which detects clusters of alleles at similar frequencies around a balanced locus as previously described in ref. 62. For this, the folded SNP frequencies were determined using VCFtools and fed to BetaScan with a -fold flag, retaining positive values for 9,800 SNPs in 1,382 genes.

We next applied the integrated haplotype score (iHS) statistic, for detecting evidence of recent positive directional selection from extended haplotype homozygosty calculated with the Rehh R package[63]. This iHS analysis focused on 12,416 SNPs with a population-wide minor allele frequency of at least 0.03. Names and summary annotations of gene coding sequences refer to the *P. falciparum* 3D7 strain reference as presented in the PlasmoDB browser (www.plasmodb.org)[64].

## Identifying *P. falciparum* genomic changes occurring in The Gambia after 1966-1971

To identify genomic regions differentiated between 1966–1971 and 2015 in the Gambia, 12,416 common biallelic SNPs, the same set used for the iHS analysis noted above, were employed to conduct a scan using 'diff_stat' function with mmod package in R. The $F_{ST}$ index was calculated for each SNP as Nei's $G_{ST}$ and visualised on a genome-wide Manhattan plot using package qqman in R. Regions of SNPs in genes with highly differentiating missense (non-synonymous) variants were further visualised with zoomed in $F_{ST}$ scatter plots highlighting the variants located in coding sequences in PlasmoDB[64]. Customised script with TopR and locuszoom package in R, were used applied to reveal details around differentiating loci. To extend the temporal analysis to include allele frequencies at intervening times over the period, previously obtained *P. falciparum* genomic sequence data from population samples taken in 1984 and 2001 were analysed[1].

## Data availability

All original data reported here are fully available as outlined in the tables and data spreadsheets in the Supplementary Data 1–9. This includes the parasite genome sequences from each of the individual samples, for which the European Nucleotide Archive (ENA) accession numbers are listed in Supplementary Data 1 and 6.

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

## Acknowledgements

We are grateful to current and previous members of staff at the MRC Unit in The Gambia and the London School of Hygiene and Tropical Medicine who, at different times enabled the maintenance of research sample archives. We thank Lindsay Stewart for support with DNA extraction processing, as well as Eleanor Drury, Victoria Simpson and Sonia Goncalves at the Wellcome Sanger Institute for support with processing of samples for sequencing. This research was supported by a European and Developing Countries Clinical Trials Partnership (EDCTP) Senior Fellowship Plus award (TMA2019SFP-2843-EGSAT) and a Wellcome Sanger Institute Senior International Fellowship award (S4739-IF-A.A.-N.) to A.A.-N., and an MRC Project Grant (MR/S009760/1) to D.J.C.

## Author contributions

A.A.-N.: Conceptualization, Resources, Methodology, Data curation, Investigation, Formal analysis, Validation, Funding acquisition, Writing – original draft, Writing – review and editing. M.F.D.: Investigation. C.J.D.: Resources, Writing – review and editing. U.d'A.: Resources, Writing – review and editing. D.P.K.: Methodology, Resources. D.J.C.: Conceptualization, Resources, Methodology, Data curation, Investigation, Formal analysis, Validation, Funding acquisition, Writing – original draft, Writing – review and editing.

## Competing interests

The authors declare no competing interests.
