## [Transparent Peer Review file · Communications Biology]

Major features of parasite adaptation revealed by genomes of *Plasmodium falciparum* population samples archived for over 50 years

Corresponding Author: Professor David Conway

Version 0:

Reviewer comments:

Reviewer #1

(Remarks to the Author)

The authors performed a population-based genome-wide scan of 54 archived *P. falciparum* infection samples collected from the Gambia between 1966 and 1971 and identified 26 genes with strong signatures of positive selection in an era before the emergence of drug resistance. They then sequenced an additional 89 samples collected in 2015 and compared the genomic complexity of infections as well as population-wide genomic diversity. The levels of mixed infections were much lower in 2015 than in 1966-1971, suggesting population contraction likely due to drug selection. Indeed, four drug-resistance genes (*dhfr*, *aat1*, *crt*, and *dhps*) were among the genes with exceptional allele frequency changes over time. Other genes with high levels of allele frequency change over time included *pfsa1*, *gdv1*, *pfsa3*, *msh7*, *mfs6*, and *cpp-5*; however, the selection drivers or mechanisms underlying these changes are largely unknown. Studying the potential forces driving these changes would reveal some interesting biological mechanisms. The manuscript is well-written, and the observations are interesting.

I only have a couple of thoughts for the authors to comment.

Line 86-87: '14 of these genes contained SNPs with significant integrated haplotype scores (iHS), indicating recent positive directional selection on some alleles'. These 14 had been through both balancing and recent positive selections. Can the authors elaborate on how both selections occur in the context or time of selection?

Lines 104-105: A positive Tajima's D can also indicate a recent population contraction or bottleneck. Resistance to chloroquine emerged in The Gambia by the late 1980s, indicating that chloroquine had likely been used before that time. My Google searches returned the following sentence: "From 1966 to 1971 in The Gambia, chloroquine was the primary antimalarial drug used, with sulfadoxine-pyrimethamine becoming available in 1971 as a second-line therapy." In that case, there could have been some population contraction between 1966 and 1971 due to drug use. I agree that most genes are likely under balancing selection; however, some of the signals could be due to chloroquine-mediated population contraction. The authors may want to consider and discuss this possibility.

Minors:

Abstract, Understanding the evolution of human pathogens requires looking beyond the effects of recent..

Line 128, the levels of mixedness was (were).

Line 214, diversity in the the oldest archived

Line 237, key role of in regulating

Line 253, As the frequency of sickle cell genotypes have not

Line 255, variants affect the fitness of infections

Line 261, extent on the original methods

Line 332, statistic was calculated using

Table 1. The number of SNPs was provided for each gene. It would also be beneficial to provide the size of the gene (how many bp).

Fig. 1 Legend, prior to the emergence of drug resistance.

Fig. 3 legend, A wide range of between-infection genomic diversity is shown as a scatter plot.

Fig. 4 legend, as an integral membrane component

There are some more similar minor typos/issues. Please check.

Reviewer #2

(Remarks to the Author)

Using historical, archived, placental blood samples, dating to more than 50 yrs ago, collected from pregnant malarious Gambian women, the present study presents a truly fascinating and unique window into the past of a *Plasmodium falciparum* population. Analyses of complexity of infection and of distribution patterns of genome-wide genetic variants among 54 samples, collected between 1966 and 1971, revealed prevalent selective pressures on the population of *P. falciparum* in The Gambia at the time that preceded the use of antimalarial drugs. Results are consistent with the high malaria transmission intensity in the country at the time, and strong selective pressure imposed primarily by the vertebrate host's immune system on the parasite population. Comparison of historical and current sample sets revealed how the *P. falciparum* population has evolved, with major shifts in allele frequency in loci associated with drug resistance, as well as adaptation to changing transmission dynamics and host populations. While some results would have been expected (eg, evol of loci involved in resistance to antimalarial drugs), others are unexpected, but make sense in hindsight, and others still open new avenues for inquiry. Despite providing a very interesting read, a few issues require attention.

1) Historical samples were collected from placental blood while contemporaneous samples were collected from uncomplicated malaria cases. Immune system dampening during pregnancy may be associated, causally, with the higher complexity of infection (COI) observed in historical isolates. This brings into question the main source of the difference in COI between historical and contemporaneous samples. The authors address this possibility (P7L219) by referencing two papers reporting lack of difference in COI among placental, cord blood and peripheral blood from pregnant women. However, those papers did not compare samples from pregnant and non-pregnant women, which makes them inconsequential to the main point here. This important caveat needs to be properly presented and discussed.

2) The same factor could, at least in theory, impact the differentiation between loci in the two time points. It would be interesting if additional metadata associated with older and recent isolates was known (eg, are the older samples from a random collection of pregnant women or from a set of women with specific sickle cell phenotype? Are recent samples from both men and women or just one or the other?). Again, this should be discussed as a potential caveat of the study.

3) Abstract, L32 "adaptations that have not reached equilibrium": there is no evidence of whether or not loci observed evolving (changing in allele freq) between 1966 and 2015 have reached/are at equilibrium. Should be rephrased.

4) Section on 'Genomic complexity of infections...' and Fig 2. Data are not normally (or even symmetrically) distributed. Probably best to report median values. Pls explain smaller horizontal lines (inter quartile range?).

5) Section on "Population-wide genomic diversity..." and Fig 3.

a. L586/587 "MDS based on the genomic profile of each of the infection samples": this is vague. Was the MDS based on the pairwise distance matrix?

b. Not sure what information is to be extracted from fig 3A.

c. Is there a relationship between the clusters in Fig 3A and the clades in Fig 3B?

d. Fig 3B: pls make it clear that scale refers to %difference only among variable sites (not genome-wide).

e. L140: "shows a high diversity": how is that conveyed? Pls explain.

6) L211-213: The fact that more antigen-encoding genes were identified in the older sample set compared to the most recent does not necessarily mean that immune selection is now weaker. This could be an artifact of other, stronger selective pressures (imposed by antimalarials) erasing (some of) the signatures of selection imposed by the immune system.

7) Methods section

a. L299: "biallelic SNPs data": how many tri- and tetra-allelic sites were removed and could removal of these sites impact metrics such as Tajima's D, iHS or betaScan?

b. L301: "filtration to not more than 20% missingness": is this across sites (ie for sample) or across samples (ie, filtering sites) or both?

c. Confirm that Tajima's D was calculated without removing variable sites with low frequency.

Minor

8) Italicize 'D' in Tajima's D throughout

9) L128 mixedness -> complexity

10) Should apply standard usage for terms such as codon vs amino acid residue, allele vs peptide variant, locus vs. variable site throughout the text (e.g., L165: P217H is not a "codon change" but a change in amino acid residue; L165 "217H allele" -> 217H variant; L182 "Another locus" -> another polymorphic site)

11) L313-319: fix formatting of FWS

12) L319: "mpb"?

13) L344: typo highly

14) L345: typo scrit

Version 1:

Reviewer comments:

Reviewer #1

(Remarks to the Author)

Figure 3 consists of two panels, both labeled as A and B.
I have no other comment.

Reviewer #2

(Remarks to the Author)

The authors have addressed my concerns adequately.

RESPONSES TO REVIEWERS

We are grateful for the helpful and insightful comments of the reviewers, who appreciated the work and noted a number of points on which additional information and clarification of interpretation should be provided. We have made revisions accordingly, presenting new details as well as improving the presentation of information that had not been as clear as intended. We think that reflecting and responding positively to the very constructive perspectives of the reviewers has improved the manuscript.

We include a 'track changes' version of the revised manuscript, and a summary of changes in response to the individual points of the reviewers are NOTED BELOW IN CAPITALS FOR CLARITY.

Reviewer #1 (Remarks to the Author):

The authors performed a population-based genome-wide scan of 54 archived *P. falciparum* infection samples collected from the Gambia between 1966 and 1971 and identified 26 genes with strong signatures of positive selection in an era before the emergence of drug resistance. They then sequenced an additional 89 samples collected in 2015 and compared the genomic complexity of infections as well as population-wide genomic diversity. The levels of mixed infections were much lower in 2015 than in 1966-1971, suggesting population contraction likely due to drug selection. Indeed, four drug-resistance genes (*dhfr*, *aat1*, *crt*, and *dhps*) were among the genes with exceptional allele frequency changes over time. Other genes with high levels of allele frequency change over time included *pfsa1*, *gdv1*, *pfsa3*, *msp7*, *mfs6*, and *cpp-5*; however, the selection drivers or mechanisms underlying these changes are largely unknown. Studying the potential forces driving these changes would reveal some interesting biological mechanisms. The manuscript is well-written, and the observations are interesting.

I only have a couple of thoughts for the authors to comment.

Line 86-87: '14 of these genes contained SNPs with significant integrated haplotype scores (iHS), indicating recent positive directional selection on some alleles'. These 14 had been through both balancing and recent positive selections. Can the authors elaborate on how both selections occur in the context or time of selection?

THIS IS NOW DONE WITH SENTENCES ADDED IN THE RESULTS AND DISCUSSION.

Lines 104-105: A positive Tajima's D can also indicate a recent population contraction or bottleneck. Resistance to chloroquine emerged in The Gambia by the late 1980s, indicating that chloroquine had likely been used before that time. My Google searches returned the following sentence: "From 1966 to 1971 in The Gambia, chloroquine was the primary antimalarial drug used, with sulfadoxine-pyrimethamine becoming available in 1971 as a second-line therapy." In that case, there could have been some population contraction between 1966 and 1971 due to drug use. I agree that most genes are likely under balancing

selection; however, some of the signals could be due to chloroquine-mediated population contraction. The authors may want to consider and discuss this possibility.

A SENTENCE IS ADDED TO THE RESULTS TO EXPLAIN THE INTERPRETATION OF TAJIMA'S D VALUES LISTED IN TABLE 1 COMPARED TO THE GENOME-WIDE BACKGROUND AS SHOWN IN SUPPLEMENTARY FIG. 1. THE GENOME-WIDE DATA SHOWS THAT THERE WAS NO SIGNIFICANT POPULATION CONTRACTION, BUT RATHER THE OVERALL NEGATIVE VALUES ARE CONSISTENT WITH LONG TERM POPULATION EXPANSION AS DESCRIBED PREVIOUSLY.

INTERESTINGLY, THE SENTENCE REFERRED TO AS RETURNED BY GOOGLE BEGINNING "From 1966 to 1971 in The Gambia..." SEEMS AN AI-GENERATED ARTEFACT IN THE GOOGLE-SPONSORED ALGORITHM, TAKING PIECES OF TEXT FROM THIS MANUSCRIPT WHICH WAS MADE AVAILABLE ON BIORXIV AT THE TIME OF INITIAL SUBMISSION. THIS MANUSCRIPT IS THE ONLY DOCUMENT TO HAVE EVER MENTIONED 1966 AND 1971 IN RELATION TO MALARIA IN THE GAMBIA, BUT THERE WAS NO INTRODUCTION OF ANY ANTIMALARIAL DRUG IN EITHER YEAR, SO THE GOOGLE AI ALGORITHM MUST HAVE INVENTED THAT. WE HAVE NOW EDITED SENTENCES TO THE INTRODUCTION TO CLARIFY AND HOPEFULLY REDUCE THE PROBABILITY OF THAT HAPPENING AGAIN.

Minors:

Abstract, Understanding the evolution of human pathogens requires looking beyond the effects of recent. DONE - EDITED

Line 128, the levels of mixedness was (were). DONE - CORRECTED

Line 214, diversity in the the oldest archived DONE - CORRECTED

Line 237, key role of in regulating DONE - CORRECTED

Line 253, As the frequency of sickle cell genotypes have not DONE - CORRECTED

Line 255, variants affect the fitness of infections DONE - EDITED

Line 261, extent on the original methods DONE - EDITED

Line 332, statistic was calculated using DONE - CORRECTED

Table 1. The number of SNPs was provided for each gene. It would also be beneficial to provide the size of the gene (how many bp). – DONE - INSERTED

Fig. 1 Legend, prior to the emergence of drug resistance. DONE - EDITED

Fig. 3 legend, A wide range of between-infection genomic diversity is shown as a scatter plot. DONE – CORRECTED. Fig. 4 legend, as an integral membrane component DONE - CORRECTED

There are some more similar minor typos/issues. Please check. DONE

Reviewer #2 (Remarks to the Author):

Using historical, archived, placental blood samples, dating to more than 50 yrs ago, collected from pregnant malarious Gambian women, the present study presents a truly fascinating and unique window into the past of a *Plasmodium falciparum* population. Analyses of complexity of infection and of distribution patterns of genome-wide genetic variants among

54 samples, collected between 1966 and 1971, revealed prevalent selective pressures on the population of *P. falciparum* in The Gambia at the time that preceded the use of antimalarial drugs. Results are consistent with the high malaria transmission intensity in the country at the time, and strong selective pressure imposed primarily by the vertebrate host's immune system on the parasite population. Comparison of historical and current sample sets revealed how the *P. falciparum* population has evolved, with major shifts in allele frequency in loci associated with drug resistance, as well as adaptation to changing transmission dynamics and host populations. While some results would have been expected (eg, evol of loci involved in resistance to antimalarial drugs), others are unexpected, but make sense in hindsight, and others still open new avenues for inquiry. Despite providing a very interesting read, a few issues require attention.

1) Historical samples were collected from placental blood while contemporaneous samples were collected from uncomplicated malaria cases. Immune system dampening during pregnancy may be associated, causally, with the higher complexity of infection (COI) observed in historical isolates. This brings into question the main source of the difference in COI between historical and contemporaneous samples. The authors address this possibility (P7L219) by referencing two papers reporting lack of different in COI among placental, cord blood and peripheral blood from pregnant women. However, those papers did not compare samples from pregnant and non-pregnant women, which makes them inconsequential to the main point here. This important caveat needs to be properly presented and discussed.

WE AGREE AND APPRECIATE THE POINT – CHANGES HAVE BEEN MADE TO THE TEXT ACCORDINGLY.

2) The same factor could, at least in theory, impact the differentiation between loci in the two time points. It would be interesting if additional metadata associated with older and recent isolates was known (eg, are the older samples from a random collection of pregnant women or from a set of women with specific sickle cell phenotype? Are recent samples from both men and women or just one or the other?). Again, this should be discussed as a potential caveat of the study.

WE HAVE ADDED THE DETAILS TO THE METHODS SECTION, CLARIFYING THAT THERE WAS NO SELECTION OF PREGNANT WOMEN FOR ANY CONDITION, AND THAT THE MORE RECENT MALARIA CASES SAMPLED WERE FROM MALES AND FEMALE CHILDREN AGED UP TO 14 YEARS.

3) Abstract, L32 “adaptations that have not reached equilibrium”: there is no evidence of whether or not loci observed evolving (changing in allele freq) between 1966 and 2015 have reached/are at equilibrium. Should be rephrased.

WE AGREE AND HAVE REMOVED THE PHRASE ACCORDINGLY.

4) Section on ‘Genomic complexity of infections...’ and Fig 2. Data are not normally (or even symmetrically) distributed. Probably best to report median values. Pls explain smaller

horizontal lines (inter quartile range?).

WE HAVE INDICATED THE MEDIAN VALUES IN THE TEXT, WHICH WERE NOT MARKELY DIFFERENT FROM THE MEANS (E.G., 0.74 VS 0.76 FOR MEADIAN AND MEAN RESPECTIVELY FOR THE SAMPLES FROM 1966 – 1971). AND SO WE RETAINED THE LINES SHOWING THE MEANS FOR MORE COMPLETE INFORMATION. WE APPRECIATE THE LINES INDICATING RANGES (WHICH WERE STANDARD DEVIATIONS) WERE NOT NEEDED AND WE HAVE REMOVED THESE.

5) Section on “Population-wide genomic diversity...” and Fig 3.

a. L586/587 “MDS based on the genomic profile of each of the infection samples”: this is vague. Was the MDS based on the pairwise distance matrix? MORE DETAILS HAVE NOW BEEN ADDED. THE MDS WAS BASED ON PAIRWISE DISTANCES WITH A SET OF 2822 SNPS AS NOW EXPLAINED MORE CLEARLY.

b. Not sure what information is to be extracted from fig 3A. WE HAVE NOW GIVEN CLEARER INFORMATION ON THE ANALYSIS AND ON HOW THIS RELATES TO FIG 3B.

c. Is there a relationship between the clusters in Fig 3A and the clades in Fig 3B? YES. WE HAVE REFINED THE CLUSTERING APPROACH FOR THE MATRIX, AND HIGHLIGHT A CLUSTER AS SEEN IN FIG3A

d. Fig 3B: pls make it clear that scale refers to %difference only among variable sites (not genome-wide). WE HAVE NOW CLARIFIED THAT THE ANALYSIS IS BASED ON 2822 VARIABLE SNP SITES, AND RE-PRESENTED THE SCALE AS INDENTITY-BY-STATE (1 – THE SNP-BASED DISTANCE).

e. L140: “shows a high diversity”: how is that conveyed? WE SEE THAT USING THE TERM ‘HIGH DIVERSITY’ IN THIS CONTEXT IS UNCLER, AND HAVE NOW CHANGED THIS TO SIMPLY INDICATE A WIDE RANGE OF DIFFERENT GENOMIC PROFILES.

6) L211-213: The fact that more antigen-encoding genes were identified in the older sample set compared to the most recent does not necessarily mean that immune selection in now weaker. This could be an artifact of other, stronger selective pressures (imposed by antimalarials) erasing (some of) the signatures of selection imposed by the immune system.

WE AGREE AND HAVE ADDED TO THE TEXT ACCORDINGLY.

7) Methods section

a. L299: “biallelic SNPs data”: how many tri- and tetra-allelic sites were removed and could removal of these sites impact metrics such as Tajima’s D, iHS or betaScan? WE HAVE NOW GIVEN MORE INFORMATION ON THE EXACT NUMBERS OF SNPS USED FOR EACH OF THE DIFFERENT ANALYSES, IN THE RESPECTIVE SUBSECTIONS OF THE METHODS. ONLY A MINORITY OF SNP SITES CONTAIN MORE THAN TWO DETECTED ALLELES (2.4% OF THE ORIGINAL SET OF 49,565 SNPS IN THE 1966-1971 DATA), AND EXCLUSION WOULD HAVE ONLY A MINOR EFFECT ON ANY TEST. TAJIMA’S D INDEX MAY BE IMPACTED SLIGHTLY TO BE MORE CONSERVATIVE – IT MAY REDUCE THE SENSITIVITY OF DETECTING GENES WITH UNUSUALLY HIGH VALUES. THE PROCESS USED WAS SIMILAR IN PREVIOUS STUDIES OF GAMBIAN POPULATION SAMPLES REFERRED TO HERE (E.G. REFS 1 AND 12).

b. L301: “filtration to not more than 20% missingness”: is this across sites (ie for sample) or across samples (ie, filtering sites) or both? THIS IS ACROSS BOTH SNPS AND SAMPLES. THE TEXT HAS BEEN CORRECTED TO CLARIFY.

c. Confirm that Tajima’s D was calculated without removing variable sites with low frequency. YES, WE AGREE THIS IS IMPORTANT TO CLARIFY. ALL VARIANTS WERE USED WITHOUT REMOVING ANY WITH LOW FREQUENCIES, AND THIS IS NOW STATED IN THE METHODS.

Minor

8) Italicize ‘D’ in Tajima’s D throughout - DONE

9) L128 mixedness -> complexity - DONE

10) Should apply standard usage for terms such as codon vs amino acid residue, allele vs peptide variant, locus vs. variable site throughout the text (e.g., L165: P217H is not a “codon change” but a change in amino acid residue; L165 “217H allele” -> 217H variant; L182 “Another locus” -> another polymorphic site) - DONE

11) L313-319: fix formatting of FWS - DONE

12) L319: “mpb”? – THIS IS NOW DELETED AS IT WAS A LABEL INSERTED IN ERROR

13) L344: typo highlty - CORRECTED

14) L345: typo scrit - CORRECTED